# Interpretation of Heart and Lungs Sounds Acquired via Remote, Digital Auscultation Reached Fair-to-Substantial Levels of Consensus among Specialist Physicians

**DOI:** 10.3390/diagnostics13193153

**Published:** 2023-10-09

**Authors:** Diana Magor, Evgeny Berkov, Dmitry Siomin, Eli Karniel, Nir Lasman, Liat Radinsky Waldman, Irina Gringauz, Shai Stern, Reut Lerner Kassif, Galia Barkai, Hadas Lewy, Gad Segal

**Affiliations:** 1HIT—Holon Institute of Technology, Holon 5810201, Israelhadasl@hit.ac.il (H.L.); 2Internal Medicine Department C, Rabin Medical Center, Sharon Campus, Petah Tikva 494149, Israel; evgenyb@clalit.org.il (E.B.); dsiomin@clalit.org.il (D.S.); 3Internal Medicine Department B, Meir Medical Center, Kfar Saba 4428164, Israel; 4Internal Medicine Department I, Chaim Sheba Medical Center, Tel-Hashomer 5265601, Israel; nir.lasman@sheba.health.gov.il; 5Internal Medicine Department A, Chaim Sheba Medical Center, Tel-Hashomer 5265601, Israel; 6Geriatric Department C, Chaim Sheba Medical Center, Rehabilitation Hospital, Tel-Hashomer 5265601, Israel; 7The Azrieli Faculty of Medicine, Bar-Ilan University, 8 Henrietta Szold St, Safed 1311502, Israel; 8Department of Pediatric Intensive Care, The Edmond and Lily Safra Children’s Hospital, Chaim Sheba Medical Center, Tel-Hashomer 5265601, Israel; reut.kassif@sheba.health.gov.il; 9Beyond Virtual Hospital, Tele-Medicine Virtual Hospital, Chaim Sheba Medical Center, Tel-Hashomer 5265601, Israel; 10Education Authority, Chaim Sheba Medical Center, Tel-Hashomer 5265601, Israel

**Keywords:** digital auscultation, remote auscultation, telemedicine, hospital at home, Kapa coefficient, TytoCare^TM^

## Abstract

Background. Technological advancement may bridge gaps between long-practiced medical competencies and modern technologies. Such a domain is the application of digital stethoscopes used for physical examination in telemedicine. This study aimed to validate the level of consensus among physicians regarding the interpretation of remote, digital auscultation of heart and lung sounds. Methods. Seven specialist physicians considered both the technical quality and clinical interpretation of auscultation findings of pre-recorded heart and lung sounds of patients hospitalized in their homes. TytoCare^TM^ system was used as a remote, digital stethoscope. Results. In total, 140 sounds (70 heart and 70 lungs) were presented to seven specialists. The level of agreement was measured using Fleiss’ Kappa (FK) variable. Agreement relating to heart sounds reached low-to-moderate consensus: the overall technical quality (FK = 0.199), rhythm regularity (FK = 0.328), presence of murmurs (FK = 0.469), appreciation of sounds as remote (FK = 0.011), and an overall diagnosis as normal or pathologic (FK = 0.304). The interpretation of some of the lung sounds reached a higher consensus: the overall technical quality (FK = 0.169), crepitus (FK = 0.514), wheezing (FK = 0.704), bronchial sounds (FK = 0.034), and an overall diagnosis as normal or pathological (FK = 0.386). Most Fleiss’ Kappa values were in the range of “fare consensus”, while in the domains of diagnosing lung crepitus and wheezing, the values increased to the “substantial” level. Conclusions. Bio signals, as recorded auscultations of the heart and lung sounds serving the process of clinical assessment of remotely situated patients, do not achieve a high enough level of agreement between specialized physicians. These findings should serve as a catalyzer for improving the process of telemedicine-attained bio-signals and their clinical interpretation.

## 1. Introduction

### 1.1. Current Status of Hospital-at-Home Services

Home hospitalization services (also known as hospital-at-home (HAH) services), were already available prior to the start of the coronavirus pandemic; however, since then, these services have expanded world-wide and, indeed, are now considered to be one of the positive outcomes of the pandemic [1,2,3]. Within this unique realm of clinical medicine, telemedicine-controlled, hospital-at-home services have progressed, catching up with pre-existing technologies and methodologies that were already available but poorly adopted by the medical community in previous years [4]. Nevertheless, currently applied telemedicine tools in the field of HAH services are still poorly validated, and there are no accepted guidelines for their utilization in HAH clinical practices [5]. Open questions still exist regarding physicians’ roles in these services and the resultant need of these physicians to rely on telemedicine technologies.

### 1.2. Technology-Based, Telemedicine-Controlled Hospital-at-Home Services

Relying on telemedicine allows the recruitment of specialized, experienced physicians with less need for them to actually reach patients’ homes, rather staying in their own homes or clinical headquarters and managing their patients using telemedicine. The physical gap between the patient and the physician should be filled by a combination of experience and advanced technologies [6,7]. The application of technologies is intended, among other tasks, to facilitate the conduction of physical examinations.

### 1.3. The Case of Technologically Enabled Remote Physical Examination

Certain aspects of the physical examination are being omitted in the setting of telemedicine-controlled HAH [8,9]: palpation is still unavailable, and the remote physician should rely on surrogate indicators for palpation-based findings. For example, asking the patient to cough and viewing the reaction related to abdominal discomfort can help to diagnose cases of peritoneal irritation. Nevertheless, other components of physical examination, like auscultation of physiological and pathological sounds, should not be waived. With the advance and dissemination of telemedicine worldwide, digitally acquired remote auscultation of the heart and lungs became an acceptable alternative to legacy measures [10,11]. Previous publications described digital techniques developed for the computerized analysis of such remotely acquired sounds [12]. Nevertheless, the validation of such data is scarce: Javed et al. [13] tested the external validity of digitized heart sounds, combined with clinical data, by comparing the computerized analysis of sounds through the interpretation of four specialized cardiologists for a total of 40 patients. They found out that their system, designated as an “embedded intelligent heart sound analyzer”, coincided with the clinicians’ diagnoses. It is noteworthy that no previous clinical study has tested the internal validity of systems designed for the remote auscultation of patients in the hospital-at-home setting.

### 1.4. Application of the TytoCare^TM^ System in Telemedicine-Controlled HAH

TytoCare^TM^ is a digital system applied in such settings [10,14]. The TytoCare^TM^ system incorporates a digital otoscope, an infrared thermometer, a tongue depressor, and an FDA-approved digital stethoscope, enabling the remote physical examination of the ears, throat, skin, heart, and lungs. Through a touchscreen interface, users can easily perform a remote physical examination that is shared with medical providers either asynchronously or in real time. The data are transmitted through a cloud-based portal, where it can be stored and reviewed by physicians at a later date. In order to ensure the accuracy of clinical information, both clinicians and patients must utilize reliable validated diagnostic equipment and efficient methods for data collection and transfer. As demonstrated recently, TytoCare^TM^ devices produce high-quality sounds and images compared to those of other stand-alone traditional devices [14,15].

As telemedicine advances, clinicians are expected to further develop clinical scenarios for this novel realm of medicine while providing clinical validations of their application of advanced technologies [16,17,18]. These validations should be both internal (validating the usability and reliability of the tool among different users) and external (comparing novel technologies with previous legend tools of clinical application) [19,20]. The current study followed a long period, during which we used the TytoCare^TM^ system for telemedicine-controlled HAH services. This study aimed to assess the degree of consensus among specialists in internal medicine using the TytoCare^TM^ remote auscultation device.

## 2. Materials and Methods

This study was an internal validation study intended to establish the extent of agreement between specialists in internal medicine regarding the clinical interpretation of heart and lung sounds acquired in the setting of telemedicine-controlled, hospital-at-home hospitalizations. The TytoCare^TM^ system was used as a digital stethoscope by nurses in patients’ homes. The auscultations were obtained after the systematic instruction of these nurses, who recorded the heart and lung sounds and transmitted them to the attending physician in the offline mode of the TytoCare^TM^ cloud-based system. We adopted the original scenario of TytoCare^TM^, in which patients themselves were instructed to record their heart and lungs sounds and upload them onto the TytoCare^TM^ cloud. In the setting of our HAH service, we did not intend to rely on patients and families but to deliver the remote physician the best auscultation data available.

### 2.1. The TytoCare^TM^ System Auscultations

The increased use of telemedicine and virtual software worldwide offers enhanced development and usage of immediate access to digital healthcare, with remote physical examination tools being adopted [21]. As such, TytoCare^TM^ device allows patients to perform examinations and transmitting sound, video, or image data to remotely seated medical professionals. The TytoCare^TM^ Device connects to an application on the patient’s mobile device to facilitate the process of communicating examination data and performing an online meeting with the clinician [11]. TytoCare^TM^ Stethoscope use a frequency range of 20–3500 Hz, intended to measure heart rate sounds in the range of 30–250 beats per minutes (BPM). Its dimensions are 1.57 × 1.53 inch (40 × 39 mm).

In order to provide the physical examination of the heart and lung, the TytoCare^TM^ stethoscope should be held steady and with its surface flat against the skin. In the clinical scenario, the stethoscope is put over four points on the chest wall by the nurse attending the patients’ homes via the following method: (1) aortic valve point of auscultation—second right intercostal space; (2) pulmonary valve point of auscultation—second left intercostal space; (3) tricuspid valve point of auscultation—forth left intercostal space near the lower sternal border; and (4) mitral valve point of auscultation—fifth left intercostal space in the mid-clavicular line. In the current study, we compared the auscultation recorded at point number 1 for all 70 patients. This point was chosen since it is essential to appreciate aortic valve stenosis, and we wanted physicians to gain consensus on this particular finding wherever it existed. The auscultation to the lungs was performed at 8 points: 4 points of auscultation were on the anterior chest wall, and four points of auscultation were on the back. The anterior points were the anterior upper right and left lung lobe fields and the anterior middle right and middle left lung field. Posterior points included the middle left and middle right lung field and the lower left and lower right lung field. In the current study, we compared the auscultation in the lower left lung for all 70 patients. This point was chosen since pleural effusions, often accumulating in the setting of congestive heart failure, tend to appear within the right pleura cavity and potentially affect the quality of pulmonary auscultation. We did not want the presence of a pleural effusion to interfere with the clinical interpretation of the bronchial and pulmonary sounds.

### 2.2. Patient and Physician Populations

In the current study, physicians (all specialists in internal medicine, serving as attending physicians in hospital departments, with over five years of experience) were asked to listen to a total number of 140 anonymous recordings of lung and heart auscultation sounds. All recordings were made as part of the routine clinical scenario of seventy consecutive patients who were hospitalized, as acute patients, in the telemedicine-controlled, hospital-at-home service of Sheba Beyond, a virtual hospital arm of a large tertiary medical center in Israel, between February 2022 and March 2023: 70 recordings taken at the position suitable for the assessment of the aortic valve (point 1), and 70 recordings were taken at the posterior lower left lung field. All participating physicians used the same headphones (SOYOTO gaming headset SY855MV, having driver sensitivity of 110 ± 3 dB and frequency range of 20 Hz–20 KHz), as well as the same audio player software in order to avoid bias. The accompanying short vignette, including the patients’ main complaints (e.g., chest pain or cough) and main background diagnoses (e.g., congestive heart failure), were also made available to the physicians in order to evaluate their clinical interpretation, as is always carried out in clinical scenarios. While listening to the heart and lung sounds, physicians were asked to complete an online Likert-based questionnaire regarding the interpretation of sounds related to their technical integrity, type of diagnosis, and overall quality. This questionnaire was separately completed for each sound. The Likert-based questionnaires were designed to be simple and informative, following the relevant principles recommended in the literature [12]. Exemplary questionnaires are presented in Table 1 (heart sounds) and Table 2 (lungs sounds).

### 2.3. Statistical Analysis

This study was an internal validation study in which we compared physicians’ answers to the aforementioned questionnaires with intent to measure the level of consensus between experts. The degree of agreement between experts was assessed using the Cohen’s Kappa coefficient, with levels of agreement spanning from poor to perfect (Table 3) [13]. Nevertheless, Cohen’s Kappa is commonly only used to measure agreement between two raters. Since the current study included seven independent raters, we used Fleiss’ Kappa measure of agreement, especially suitable for cases in which there are more than two raters, assessing the level of agreement above the values expected by chance. SPSS software was used for statistical analysis (IBM SPSS Statistics for windows, version 28.0.0.0).

## 3. Results

Between February 2022 and March 2023, a total of 70 consecutive patients were remotely examined by a specialist physician as part of their stay in a telemedicine-controlled, hospital-at-home service delivered by Sheba Beyond, the first virtual hospital in Israel. All sounds were routinely uploaded to a cloud portal and served the clinical purposes of diagnosis and follow-up.

The retrospective, research-based analysis of the recorded sounds only took place after the Chaim Sheba Institutional Review Board approved the study (approval number SMC-22-9739, signed at 22 November 2022). Informed consent was waived by the review board due to the retrospective nature of this study and the fact that all patients’ data were anonymized. A total 140 sounds were included in this study: heart sounds recorded at the preferred point for assessment of the aortic valve (second right intercostal space, point 1) and lung sounds derived from the posterior lower left lung field.

Seven senior physicians, all specialists in internal medicine, serving as attending physicians in their departments for more than five years, were asked to interpret the heart and lung sounds after being presented with a short written summary related to the patients’ main complaints and background diagnoses. The questions related to the recorded sounds targeted their overall quality, their compatibility with the interpretation aimed at clinical diagnosis and several questions related to the existence or absence of specified pathologies, like heart murmurs or bronchial wheezing. All participating specialists were blinded to the identities and other potentially identifying details of the patients. They separately listened to and interpreted the recorded sounds and were totally blinded to each other’s interpretations. The physicians were recruited as part of the study research team and, therefore, were not required to sign informed consent/were waived by the institutional review board.

Overall, we compared and assessed the level of consensus that would be above the expected by chance level between the seven sets of questionnaires. We used Fleiss’ Kappa measure of agreement coefficient for this purpose. The level of agreement between specialists was measured using Fleiss’ Kappa (FK) variable for the following parameters: Heart sounds (Figure 1)—the overall technical quality (FK = 0.199), existence of rhythm irregularity (FK = 0.328), presence of murmurs (FK = 0.469), appreciation of heart sounds as pathologically remote (FK = 0.011), and overall diagnosis of the examination as normal or pathologic (FK = 0.304). Lung sounds (Figure 2)—the overall technical quality (FK = 0.169), existence of crepitus (FK = 0.514), existence of wheezing (FK = 0.704), existence of bronchial sounds (FK = 0.034), and overall diagnosis of the examination as normal or pathologic (FK = 0.386). Most Fleiss’ Kappa values were in the range of “fare consensus”, while in the domains of diagnosing lung crepitus and wheezing, the values increased to the “substantial” level of consensus.

## 4. Discussion

The field of telemedicine is undergoing rapid evolution, expanding into new realms [22]. Worldwide, guidelines for the telemedical management of remote patients are being regularly published, guiding clinicians in managing two distinct populations of patients. The first group consists of those who suffer from chronic diseases, for whom telemedicine serves as an alternative to outpatient clinics visits. The second group includes patients suffering from acute illnesses, for whom telemedicine serves as an enabling technology for the hospital-at-home service, replacing their stay in hospital, mainly in internal medicine departments.

Sheba Beyond, an Israeli virtual hospital, is unique in terms of applying telemedicine technologies and methodologies in the setting of acute, home-based hospitalizations. To date, no previous publications have assessed this innovative, clinical approach. The Sheba Beyond methodology for hospital-at-home care relies on senior, experienced physicians who remotely follow and treat their patients via telemedicine, while nurses make in-home visits to the patients. This methodology enables us to treat patients with acute illnesses otherwise needing to be hospitalized. The TytoCare^TM^ telemedicine platform, comprising a digital stethoscope and application, serve as a pillar in this setting, enabling the execution of a remote physical examination by our physicians. This methodology is new and continuously escorted by an array of clinical studies aiming to validate both the methods and applications used. The current study was designed in order to provide internal validation for the use of the TytoCare^TM^ technology within this specific setting. The researchers and physicians participating in this study represent the milieu of our practicing physicians: all are specialists in internal medicine who practice clinical medicine of acute patients on a daily basis. We believe that such physicians are well able to deliver high-quality treatment to acute patients through telemedicine channels, eliminating the need for in-home visits.

Surprisingly, most auscultation recordings’ parameters reached lower than expected levels of consensus in our population of physicians. A previous publication that demonstrated strong agreement levels between teleauscultation and traditional stethoscope-appreciated murmur detection was limited to specialized cardiologists [23]. The lowest consensus was achieved for appreciation of pathologically remote heart sounds (potentially representing the effects of a pericardial effusion) and the recognition of bronchial lung sounds (potentially representing the presence of intra-bronchial secretions). The highest levels of agreement were achieved for the diagnosis of bronchial wheezing and identifying crepitus over the lungs. We can only speculate that these sounds were recognized more easily since these are both high-pitched sounds, whereas remote heart sounds and bronchial lung sounds are both low-pitched sounds.

To the best of our knowledge, no previous comparison with simultaneous review by multiple physicians has previously been performed. Additionally, we are not aware of any external validation conducted in similar clinical scenarios that compared a full array of auscultation findings of both heart and lungs. We assume that similar results would have been reached through the use of non-digital tools.

Our results suggest three optional strategies in order to improve the quality of physical examinations performed via remote–digital auscultation as part of a telemedicine-controlled HAH setting:Recording of auscultations should follow a standardized procedure: In contrast to the traditional physical examination conducted by a physician or a nurse on their own, the attainment of digital bio-signals should be standardized. We approached this issue by dictating the specific points of auscultation performed via TytoCare^TM^ in specific patients’ positioning. This step alone did not achieve a high enough level of agreement among the interpreting physicians. Therefore, we should further standardize the process of recording auscultations related to the positioning of patients, prior training of physicians in the usage of the specific digital device used, the length of recording, controlling the level of surrounding noises, and determining the minimal number of recordings per auscultation point.Digital data related to physical examination should be interpreted by more than one healthcare professional: Currently, recorded auscultations are uploaded to the TytoCare^TM^ cloud, allowing multiple professionals to give their interpretations. These professionals can include two physicians or a combination of physicians, experienced nurses, and paramedics. The specific combination should be separately sought by each healthcare organization utilizing such technologies. At times, a cautious approach should prefer the opinion of a single physician rather than pursue a definitive solution.Artificial intelligence applications potentially interpreting bio-signals of telemedicine-gained physical examination outputs should be developed: With advancements in AI algorithms, we anticipate that, in the near future, AI applications will support physicians by providing suggestions for interpreting auscultation findings, among other bio-signals recorded during telemedicine-based physical examination of remotely situated patients [24]. When related to recommendation A, it is advisable to remember that AI algorithms will, initially, offer standardizations for acquired sounds prior to human interpretation and potentially triage sounds, flagging them as “normal” versus “necessitating prompt clinical interpretation”.

## 5. Conclusions

In the near future, telemedicine-based, hospital-at-home settings will become a global norm. Bio signals, including recorded auscultations of heart and lung sounds, will continue to serve as a pillar of the process of clinical assessment of remotely situated patients. Currently, the appreciation of such recordings does not achieve a high enough level of agreement between specialized physicians. These findings should prompt the standardization of such methodologies. We suggest considering appreciation by more than one healthcare professional and the development of AI-based applications to assist physicians in the standardization and interpretation of bio-signals.

## Figures and Tables

**Figure 1 diagnostics-13-03153-f001:**
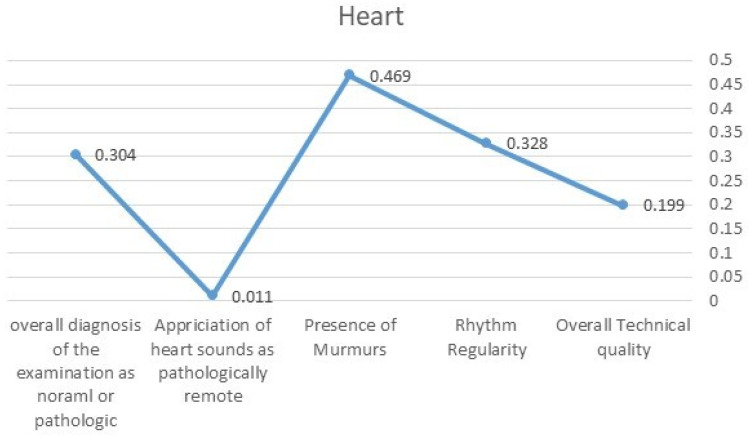
FK values demonstrating the level of agreement related to heart sounds.

**Figure 2 diagnostics-13-03153-f002:**
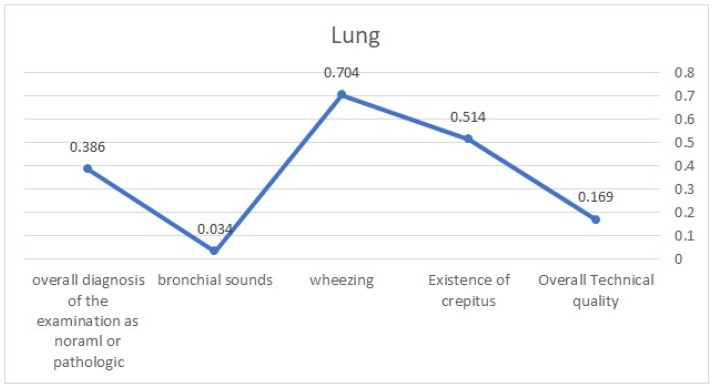
FK values demonstrating the level of agreement related to lung sounds.

**Table 1 diagnostics-13-03153-t001:** Exemplary questionnaire for heart sounds interpretation.

**Main complaint**	(e.g., chest pain)
**Background diagnoses**	(e.g., congestive heart failure)
**Technical quality**	high/low/does not enable interpretation
**Crepitus with or without crackles**	Yes/No
**Wheezing**	Yes/No
**Bronchial sounds**	Yes/No
**Summary**	normal/pathologic/does not enable interpretation

**Table 2 diagnostics-13-03153-t002:** Exemplary questionnaire for lung sounds interpretation.

**Main complaint**	(e.g., cough)
**Background diagnoses**	(e.g., congestive heart failure)
**Technical quality**	high/low/does not enable interpretation
**Regular**	Yes/No
**Murmurs**	Yes/No
**Remote sounds**	Yes/No
**Summary**	normal/pathologic/does not enable interpretation

**Table 3 diagnostics-13-03153-t003:** Kappa coefficient degree of agreement.

κ	Interpretation
<0	Poor agreement
0.01–0.20	Slight agreement
0.21–0.40	Fair agreement
0.41–0.60	Moderate agreement
0.61–0.80	Substantial agreement
0.81–1.00	Almost perfect agreement

## Data Availability

This study data is available with the corresponding author upon request.

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
