# Peer review of "Interpretation of Heart and Lungs Sounds Acquired via Remote, Digital Auscultation Reached Fair-to-Substantial Levels of Consensus among Specialist Physicians"

_diagnostics, 2023, doi:10.3390/diagnostics13193153_

Round 1
Reviewer 1 Report
The primary objective of the study is somewhat ambiguous due to a discrepancy in the information presented. Although the abstract states that the consensus level will be measured, the text suggests that the study aims to determine additional clinical applications of the TytoCareTM remote auscultation device by measuring the level of consensus among experts in Internal Medicine.
The manuscript would benefit from a more thorough literature review, as the current one falls short. Additionally, it is worth noting that the author's claim that no previous studies addressed the same objectives is inaccurate. Improvements to the manuscript's structure are also necessary, particularly in the "patient and method" section, which could benefit from additional clarity.
The authors recommended including the advice of two physicians or one physician and a skilled nurse as part of their suggested strategies. Unfortunately, the study revealed a lack of agreement among medical professionals, casting doubt on the authors' inference. Consequently, it would be prudent to adjust the recommendation to a more cautious approach, relying on the opinion of a single doctor rather than a definitive solution.
AI's potential is immense, generating a lot of enthusiasm. Nevertheless, the authors should furnish more details regarding how their suggested AI solution can proficiently tackle the issue of precise sound interpretation. It would be advantageous to lucidly exhibit the connection between the problem and the suggested solution.
The manuscript necessitates significant revisions and meticulous proofreading to ensure that it meets the requisite standards of quality.
Author Response
We thank you very much for your contributing comments to our manuscript and for the improvement of its quality.

Reviewer 2 Report
Dear Authors,
Thank you very much for your well-written manuscript dealing with an interesting issue. Please find below my comments and questions, pertaining to your manuscript:
1. According to the lines 125-127: In the current study the auscultation was compared in the lower left lung for all 70 patients. This point was chosen since pleural effusions, often accumulating in the setting of congestive heart failure, tend to appear within the right pleura cavity, potentially affecting the quality of pulmonary auscultation. According to the lines 188-189 lung sounds were collected from the posterior lower right lung field. Please make this point clear.
2. Line 190: please define the specialty and the experience of senior physicians (internal medicine specialists, cardiologists, pulmonologists?) and how many years of clinical experience (in inpatient setting, in ambulatory setting, over 5 years’ experience?).
3. Figure 1 and Figure 2: please correct as such: overall diagnosis of the examination as normal or pathologic.
4. Were the measures done in the acute clinical setting (symptomatic patients) or as a part of a follow-up examination in a regular home based nurse visit?
Best Regards
Minor editing of English language required
Author Response

(The authors gave the same response as above.)

Round 2
Reviewer 1 Report
The manuscript has improved significantly through the authors' efforts to refine the study's aim and restructure the text. Furthermore, I suggest additional enhancements to the recommendations.
Investing extra time in proofreading the manuscript could significantly enhance its overall quality.
Author Response
On behalf of all researchers, I thank you for reviewing and improving our manuscript.
Prof. Gad Segal, MD

Reviewer 2 Report
Dear Authors,
Please pay attention to the following questions and queries, pertaining to your manuscript:
1. Line 65: please correct as such: to facilitate the conduction of physical examination.
2. According to line 145: you compared the auscultation in the lower left lung for all 70 patients. On the other hand, according to line 159: 70 recordings were taken from the posterior lower right lung field. Where was the recording tested? In the lower left lung or in the lower right lung field? Please correct accordingly.
3. Line 164. “in order to evaluate their clinical interpretation as it would have been made in real-cases scenarios”: but these were all real-cases scenarios, as your patients were suffering an acute illness (according to line 155) and were hospitalized in the telemedicine-controlled, hospital-at-home service. Please revise this point.
4. Line 152. please correct as such: with over five years of experience.
5. Line 220: Please revise according to the second reviewer comment (left or right lower lung field?).
6. Lines 295-296: please revise as such: To our best knowledge, no previous comparison with simultaneous review by multiple physicians has been performed before.
7. Lines 322-323: please correct as such: we anticipate that, in the near future, AI applications will support physicians by providing suggestions.
Best Regards
Minor changes required, please find my suggestions under the reviewer comments.
Author Response

(The authors gave the same response as above.)
